# When Atrial Fibrillation Meets Cerebral Amyloid Angiopathy: Current Evidence and Strategies

**DOI:** 10.3390/jcm12247704

**Published:** 2023-12-15

**Authors:** Pierluigi Merella, Gavino Casu, Paola Chessa, Enrico Atzori, Stefano Bandino, Gianluca Deiana

**Affiliations:** 1Department of Cardiology, Azienda Ospedaliero Universitaria di Sassari, Via De Nicola 1, 07100 Sassari, Italy; gavino.casu@aouss.it (G.C.); enrico.atzori@aouss.it (E.A.); stefano.bandino@aouss.it (S.B.); 2Faculty of Medicine, University of Sassari, 07100 Sassari, Italy; 3Department of Pharmacy, San Francesco Hospital, 08100 Nuoro, Italy; paola.chessa@aslnuoro.it; 4Department of Neurology and Stroke Unit, San Francesco Hospital, 08100 Nuoro, Italy; gianluca.deiana@alsnuoro.it

**Keywords:** cerebral amyloid angiopathy, atrial fibrillation, left-atrial appendage occlusion, high bleeding risk, intracranial hemorrhage, anticoagulation

## Abstract

Non-valvular atrial fibrillation (AF) and cerebral amyloid angiopathy (CAA) are two common diseases in elderly populations. Despite the effectiveness of oral anticoagulant therapy in cardioembolic stroke prevention, intracranial hemorrhage represents the most serious complication of these therapies. Cerebral amyloid angiopathy is one of the main risk factors for spontaneous intracranial bleeding, and this risk is highly increased by age and concomitant antithrombotic therapies. Cerebral amyloid angiopathy can be silent for years and then manifest with clinical features simulating TIA (TIA-mimics) or stroke in AF patients, pushing clinicians to rapidly start VKAs or DOACs, thus increasing the risk of intracranial bleeding if the diagnosis of CAA was unknown. Because the cerebral amyloid angiopathy is easily diagnosed with non-contrast MRI, suspecting the disease can avoid catastrophic complications. In this review, we will provide physicians managing anticoagulant therapies with key tips to familiarize themselves with cerebral amyloid angiopathy, with a focus on the possible clinical presentations and on the diagnostic criteria.

## 1. Introduction

Non-valvular atrial fibrillation (AF) is the most common cardiac arrhythmia and its prevalence is growing as consequence of aging [1]. In Western countries, the lifetime risk of developing AF is about 15% [2], and 30% of all ischemic strokes are due to AF [3].

Vitamin K antagonists (VKAs) have shown their ability to decrease ischemic stroke by about 60%; these results were obtained with an increased rate of major bleeding [4]. In the last decade, direct oral anticoagulants (DOACs) have shown a similar efficacy, but with a lower risk of intracranial hemorrhage (ICH) when compared to VKAs [5,6,7,8].

Intracranial hemorrhage (ICH) represents the most serious complication of anticoagulant therapy, despite the therapy’s effectiveness.

Cerebral amyloid angiopathy (CAA) is one of the main risk factors for spontaneous ICH [9], and this risk is highly increased by age and antithrombotic therapies [10]. Moreover, CAA can be manifested with clinical features simulating TIA or stroke in AF patients [10], pushing clinicians to rapidly start VKAs or DOACs, increasing the risk of ICH if the diagnosis of CAA is unknown.

In this review, we will provide physicians managing anticoagulant therapies with key tips to familiarize themselves with CAA and avoid the risk of unnecessary ICH in patients with AF.

### 1.1. Cerebral Amyloid Angiopathy

CAA is a common cerebral small-vessel disease in elderly populations, characterized by the deposition of β-amyloid protein in the walls of arteries of small-to-medium size, arterioles, and capillaries. CAA usually affects leptomeningeal and cortical cerebral vessels [11,12].

In the first stages of the disease, β-amyloid accumulates in the basal lamina of tunica media, replacing smooth muscle cells and connective tissue [13]. In the late stages, the arterial wall is completely replaced by β-amyloid, and this may be associated with fibrinoid necrosis, cleavage between the tunica media and tunica intima, with microaneurysms formation [13,14].

Vascular alterations expose to spontaneous vascular rupture, which often occurs in normotensive elderly patients [15].

### 1.2. Epidemiology of CAA

Aging and Alzheimer’s disease (AD) are definite risk factors for CAA [16].

CAA has a prevalence of 5–9% in men between 60 and 69 years, rising up to 43–58% in patients over 90 years [17]. A recent meta-analysis reported moderate-to-severe CAA in a striking 23% of the general population (which included patients with stroke and/or dementia) and in 6% of cognitively normal elderly people, with a similar mean age for the two populations [18]. The same meta-analysis stated the presence of asymptomatic strictly lobar cerebral microbleeds, a typical radiological marker of CAA, in 7% of cognitively normal elderly people [18].

Although CAA is generally a sporadic disease, autosomal dominant forms were reported [19,20,21,22]. ApoE polymorphisms were reported to be a risk factor for sporadic cases of both CAA and AD [21].

There are also non-genetic risk factors. The prevalence of hypertension has been reported to be higher in patients with CAA, and hypertension could contribute to the progression of CAA-related vasculopathy and, in particular, to fibrinoid necrosis, which is strongly associated with the development of intracranial hemorrhage (ICH) [23]. Moreover, arterial hypertension has a well-defined role as a trigger for CAA-associated clinical syndromes [24]. Of interest, various authors have demonstrated the existence of an association between CAA and cardiac damage, as occurs, for example, in some conditions of cardiac amyloidosis [25].

## 2. CAA Clinical Presentation

### 2.1. CAA Is Generally Asymptomatic for Several Years

The main manifestations are lobar cerebral hemorrhage, cognitive decline, transient focal neurological episodes (TFNEs) and pseudotumoral presentations (Table 1).

Lobar cerebral hemorrhages are often accompanied by headache, focal neurological signs, seizures, or alterations in the state of consciousness [26].

CAA is usually associated with lobar ICH, but not with deep cerebral or cerebellar ICH (Figure 1), which, instead, are more typical of hypertensive microangiopathy [9,27] (Figure 2).

Cognitive decline is another clinical presentation and it may overlap with Alzheimer’s disease [28]. CAA-related cognitive impairment usually precedes the occurrence of ICH [29].

TNFEs are usually associated with subarachnoid hemorrhages (SAHs) at the cerebral convexity [15,28,30]. These events are described as recurrent and stereotyped episodes of sensory, visual, motor, or phasic focal neurological deficits. They are often indistinguishable from a transient ischemic attack (TIA) and, therefore, they are also defined as TIA-LIKE [28,31] or TIA-MIMIC [29].

TNFEs are usually associated with acute subarachnoid hemorrhages (SAHs) at the cerebral convexity [15,28,30,32], especially in the central sulcus [33] or sub-acute/chronic cortical bleeding named cortical superficial siderosis (cSS) [34]. Their prevalence spreads from 14% of patients diagnosed with probable CAA in a multicenter cohort of 172 patients [35] to 48% of a recent review and meta-analysis [36], representing, therefore, a major clinical differential diagnosis.

Noteworthily, TFNEs often occur in patients without pre-existing cognitive deficits [37].

Finally, other forms are associated with an intense focal cerebral inflammatory reaction (Figure 3).

These are known as “Cerebral Amyloid Angiopathy-Related Inflammation”, and are characterized by pseudotumoral vasogenic edema [15] or pseudoarteritic vascular alterations, also named inflammatory angiitis linked to beta-amyloid [14,38].

### 2.2. Treatment/Management

Currently, no specific treatment can change the course of the disease. Consequent management is often based on the presenting symptoms [39]. The acute management of patients with CAA-related ICH is similar to other causes of spontaneous ICH. Attention to blood pressure and intracranial pressure is essential. When surgical intervention is performed, the mortality risk is largely unchanged compared to other types of ICH. Intraventricular hemorrhage and an age greater than 75 are associated with a worse prognosis. Moreover, CAA-related ICH is frequently recurrent. Because of this high rate, clinicians typically avoid antiplatelet agents and anticoagulants when a strong indication for anticoagulation is not present [39]. In the PROGRESS trial, blood pressure control has been associated with mortality benefits, even though CAA does not seem to be largely driven by hypertension, with a risk reduction of 77% for CAA-related ICH when blood pressure was controlled [40]. For those presenting with inflammatory forms of CAA, the optimal treatment remains to be defined [41]. Rapid clinical and radiologic responses have been reported using steroids alone or in conjunction with other immunosuppressive drugs, such as cyclophosphamide [15]. Other immunosuppressive medications have also been associated with benefits, including mofetil, mycophenolate, and methotrexate [42].

### 2.3. Diagnosis of CAA

As stated up above, CAA can behave as a *Great Pretender*, simulating Alzheimer disease, vascular dementia, TIA or stroke, brain cancer, or brain vasculitis. The clinical features of CAA are, therefore, misleading and may push clinicians toward false diagnoses.

According to Boston Criteria 2.0, CAA should be suspected primarily in patients older than 55 years with a history of single or recurrent lobar hemorrhage and/or superficial siderosis without alternative explanation [43].

Presentation with cognitive decline or dementia is frequent in elderly people but it is not specific. Patients with gradual decline are more likely associated with microhemorrhages, lobar lacunas, microinfarcts, and chronic leukoencephalopathy [39]. Stepwise cognitive decline is linked to single or recurrent lobar ICH, while rapidly progressive decline should raise suspicion for cerebral-amyloid-angiopathy-related inflammation [15].

The suspicion of TFNEs should arise from recurrent and stereotyped neurological deficits in elderly people, including transient episodes of smoothly spreading paresthesia, numbness or weakness, typically lasting from seconds to minutes, and usually resolving over a similar period [18].

A non-enhanced brain CT scan (NECT) is often carried out on suspicion of dementia or transient neurological deficits. In CAA patients, NECT shows a prevalence of moderate–severe Fazekas score leukopathy in 53% of subjects [44], a high grade of peri-vascular spaces (PVSs) located in the centrum semiovale and basal ganglia in 56%, and lacunar infarcts in 30%. Notably, these features are relevant but nonspecific, being commonly related to age, hypertension, and diabetes [45]. Consequentially, a CT scan is simply not useful in the diagnosis of CAA.

A brain MRI is superior to NECT in detecting typical imaging markers of CAA, such as subcortical cerebral microbleeds (CMBs), superficial siderosis, convexity SAH, and lobar hemorrhages [36] (Figure 1 and Figure 2). Gradient-echo T2* sequences and susceptibility-weighted sequences are fundamental in MRI protocols to achieve adequate diagnosis; their absence can lead to misdiagnoses, so they are recommended in the study of cognitive impairment and for the proper differentiation of brain mass effect [15].

Although the definite diagnosis of CAA is autoptic, clinical criteria have been validated and can assist the clinician in diagnosis. The new released Boston criteria 2.0 were recently proposed in 2022 to better include leptomeningeal and white matter characteristics in the diagnoses of probable and possible CAA [43]. The Boston criteria 2.0 consist of combined clinical, imaging, and pathological parameters (Table 2). It is important to note that a CAA diagnosis is an exclusion diagnosis.

### 2.4. Differential Diagnosis (Table 3)

Neuroimaging is fundamental to tailoring diagnosis. Gradient-echo (GE) T2*-weighted MRI and susceptibility-weighted (SW) MRI enable the highly accurate detection of CMBs, superficial siderosis, and small superficial hematomas that are hardly detectable using conventional NECT and MRI. However, these small round or linear T2* and SW hypointense lesions, pathologically representing the focal hemosiderin deposition associated with previous microhemorrhages, are also described in multiple conditions. The clinical context and the patient’s history could lead to the correct diagnosis.
jcm-12-07704-t003_Table 3Table 3Differential diagnoses of CAA.Clinical ConditionMain Radiological CharacteristicsHypertensive microangiopathyHemorrhages are typically located in the basal ganglia, in the bridge, and in the cerebellum; they are not associated with subarachnoid hemorrhage or superficial siderosis.Infective endocarditisCombination of acute SAH, cSS, CMBs, cortical/sub cortical hematomas, cortical/subcortical abscess, small pial enhancement on post-contrast imaging, ischemic stroke, or focal stenosis on vascular exploration. Note that radiological features can rapidly change under antibiotics.Heart surgery/left-ventricle assist devicesCombination of acute SAH, cSS, parenchymal hematomas, CMBs, cerebral atrophy, or white matter leucopathy.Multiple cavernomatosisThe lesions have a random distribution and size, although the classic cavernous malformations are not distinguishable from CAA-related brain microhemorrhages. Often, the lesions have the typical “popcorn” appearance.Hemorrhagic metastasesLesions have a variable size and can often be larger than micro-hemorrhages. They present contrast media enhancement.Widespread axonal damageLesions are typically located at the junction of the gray-white substance, in the corpus callosum and, in the most serious cases, in the brain stem.NeurocysticercosisCalcific nodular component visible on CT or MRI in susceptibility-weighted images, random distribution.Fat embolism syndromeIt has a typical “starfield” presentation.Lesions also show limited diffusion in diffusion MRI sequences. Radiation-induced vasculopathyMicroemorrhagic foci have a very similar appearance to CAA, but with a distribution limited to the treated area.Abbreviations: SAH, subarachnoid hemorrhage CAA; cSS, cortical superficial siderosis; CMBs, cerebral microbleeds; CAA, cerebral amyloid angiopathy; CT, computed tomography; MRI, magnetic resonance imaging.

A major cause of cerebral hematoma results from bleeding-prone small vessel diseases in hypertensive arteriopathy (including lipohyalinosis and arteriolosclerosis). In these patients, microbleeds are often localized in deep basal ganglia or cerebellar nuclei (Figure 2).

Brain trauma can lead to diffuse axonal injury with a typical subcortical and corpus callosum T2* micro lesion pattern [46].

Hemodialysis patients have both a high prevalence of CMBs, of 19.3–35%, and much higher incidence of strokes (particularly ICH), than the general population [47].

Moreover, intravascular lymphoma can mimic inflammatory CAA [48].

Other important differential diagnoses mimicking CMBs on gradient-echo T2*-weighted MRI and SW MRI are hemorrhagic metastases, multiple cerebral cavernous malformations, radiation-induced vasculopathy, neurocysticercosis [39], microsusceptibility changes on a brain MRI in critically ill patients on mechanical ventilation/oxygenation [49], and brain microbleeds after orthotopic liver transplantation [50].

Remarkably, in critical cardiology care settings, neurological complications mimicking CAA are frequent but generally oblivious. Infective endocarditis is a frequent cause of septic-embolus-related CMBs, superficial siderosis, and lobar hematomas [51] (Figure 4).

In this setting, NECT rapidly excludes large hemorrhages in patients with infective endocarditis, but MRI accurately distinguishes the whole spectrum of brain lesions, including small ischemic lesions, microbleeds, superficial siderosis, and microabscesses [52]. Finally, cerebral microbleeds are common in patients with left-ventricular assist devices [53,54], after heart surgery [55], or after tacrolimus treatment for lung transplantation [56].

### 2.5. Implications of a Missed Diagnosis

Although the diagnosis is quite simple in typical cases of lobar ICH, the major difficulties concern patients with cognitive impairment or TFNEs (Figure 1 and Figure 2). Indeed, TFNE patients have symptoms almost indistinguishable from TIA and often have other confounding comorbidities (AF, atherosclerosis, arterial hypertension, diabetes mellitus). Furthermore, a brain CT has a low sensitivity for detecting the typical elements of CAA, such as subcortical microbleeds, superficial siderosis, and chronic hemosiderin stores [57]. SAH is visible on a brain CT only in the first days after its appearance. A brain MRI is extremely sensitive and specific [57], but rarely prescribed on suspicion of TIA in real-life settings.

In patients with TFNE and concomitant AF, a misdiagnosis can have catastrophic consequences [24,58] (Figure 1). Indeed, studies have suggested a significant increase in the risk of spontaneous ICH within weeks or months of the clinical onset of CAA [24], with potentially devastating consequences when anticoagulant therapy is started. Even in patients with previous atraumatic lobar ICH, the risk of spontaneous hemorrhagic recurrence is particularly high [59], and this risk becomes prohibitive in patients exposed to oral anticoagulant therapy (OAT) [60].

## 3. Managing Atrial Fibrillation in CAA Patients

As the cardioembolic risk is related to age, the probability of meeting a patient affected by CAA and AF is high. Moreover, the risk of ICH increases exponentially with age, and this risk is higher in patients with previous atraumatic lobar hemorrhage [61,62,63]. Antithrombotic drugs in patients with CAA increase the risk of disabling ICH [12]. TFNE presentations are associated with a marked increase in the spontaneous short-term bleeding risk [24]. Several authors suggest that such patients should not be exposed to antithrombotic drugs [24].

In patients with unexplained cognitive decline or with a suspected TFNE or a previous ICH without neuroradiological data, it could be useful to obtain a brain MRI before starting OAT [64] with an appropriate MRI protocol including T2* gradient echo (GE) sequences and susceptibility-weighted (SW) sequences [15].

Without a clinical suspicion of CAA, there are no data to support the routine use of a brain MRI as a screening tool.

In patients who have developed a cerebral hemorrhage and have a coexistence of AF and CAA, the risk of ischemic stroke must necessarily be matched with the risk of hemorrhagic recurrence, which is obviously increased by exposition to OAT [65,66].

A large meta-analysis showed how the risk of hemorrhagic recurrence was markedly increased in patients with previous lobar hemorrhages compared to patients with non-lobar hemorrhages [66].

For these reasons, European guidelines recommend not to start or resume in patients with previous lobar hemorrhage and probable or definite CAA [3].

### 3.1. AF, CAA, and Anticoagulants

Assuming CAA and AF as frequent comorbidities in elderly patients, clinicians face frequent therapeutic dilemmas on the risk–benefit ratio of long-term anticoagulation. These patients have both a risk of cardioembolic complications and a risk of cerebral hemorrhage from cerebral amyloid angiopathy. Since there is no therapeutic consensus, the best therapeutic strategy should be discussed regarding the baseline risk of intracerebral hemorrhage without anticoagulation, the risk of ischemic stroke without anticoagulation, the expected increase in intracerebral hemorrhage with anticoagulation and, finally, the expected reduction in ischemic stroke risk with anticoagulation [67]. The risk of intracerebral hemorrhage varies according to the cerebral amyloid angiopathy phenotype. Patients with transient neurological episodes or cortical superficial siderosis have the highest risk of intracerebral hemorrhage.

Of importance, The HAS-BLED (hypertension, abnormal renal/liver function, stroke, bleeding history or predisposition, labile international normalized ratio, elderly, drugs/alcohol concomitantly) score is not validated for use in CAA, and so risk stratification must be based on the recurrence rates balanced against the CHA2DS2-VASc score.

Direct oral anticoagulants (DOACs) should be used preferentially to vitamin K antagonists, as the risk of intracerebral hemorrhage is lower with DOACs. Indeed, despite having a substantially overlapping efficacy in the prevention of cardio-embolic events, they presented a lower rate of bleeding [5,6,7,8], with a significant reduction in ICH rates especially [68]. For these reasons, in patients at increased risk for ICH, DOACs represent the preferential choice [68].

If it has been decided not to pursue anticoagulation, then left-atrial appendage occlusion (LAAO) should be proposed. In all cases, close blood pressure control is essential to reducing the risk of intracerebral hemorrhage [67].

### 3.2. Left-Atrial Appendage Percutaneous Occlusion

Percutaneous left-atrial appendage occlusion (LAAO) is an alternative for patients with AF who have a contraindication for long-term OAT [3,69,70,71,72,73].

LAAO has been shown to be a valid option with AF patients with high thromboembolic risk and OAT contraindication [74]. Patients with clinically manifesting CAA and AF represent ideal candidates for LAAO, as they frequently have an absolute contraindication to anticoagulant therapies [70,75].

Although there are no randomized trials, there are encouraging data on the risk/benefit ratio of LAAO in patients with previous ICH [76,77,78,79,80,81,82].

Renou studied the feasibility of LAAO in a group of patients with AF and previous ICH. In this prospective study, in which 54% of patients had a probable or possible CAA diagnosis, the authors demonstrated the safety and efficacy of LAAO followed by a single antiplatelet drug [76]. Large retrospective studies confirmed a significant reduction in the relative risk of bleeding events [79,81].

The optimal antithrombotic strategy after LAAO is a major issue. Indeed, the ICH risk appears to also be increased in patients who started antiplatelet therapy [83,84].

Although LAAO was initially followed by a brief cycle of anticoagulant therapy, over the years, this approach has changed, moving toward a “sartorial” approach, centered on the peculiarities of the patient [70]. Experimental evidence has shown how, after three months, the device surface is largely re-endothelizied [85]. Moreover, antiplatelet therapy is now largely prescribed after LAAO. In the Ewolution trial, 6% of patients did not receive antiplatelet therapy, while 7% received a single antiplatelet drug [86].

Consequently, in recent years, the use of alternative pharmacological strategies has increased, such as the reduction in the duration of the dual antiplatelet therapy (DAPT), the increasing use of the single antiplatelet therapy (SAPT) and, in patients with a prohibitive risk, the non-prescription of any antithrombotic therapy [70].

Probably, the greatest benefit of LAAO is derived from the reduced exposure timeto antithrombotic drugs.

### 3.3. Catheter Ablation of Atrial Fibrillation

AF catheter ablation represents the cornerstone of the rhythm control strategy [3]. Theoretically, maintaining sinus rhythm could improve cerebral perfusion and could reduce cognitive decline prevalence in the long term [87,88]. In selected patients, such as some CAA patients, after an effective ablation procedure, we can, at least temporarily, suspend anticoagulant therapy, therefore reducing the bleeding risk. However, this approach is only speculative at the moment.

## 4. Conclusions

CAA in patients aged over 50 years is concerning, since these patients are also at risk of atrial fibrillation and stroke. Anticoagulation is generally contraindicated in cases of CAA; thus, the identification of patients with clinically significant CAA could affect the management of atrial fibrillation.

Cerebral amyloid angiopathy is difficult to diagnose but may be suspected based on a history of lobar intracerebral hemorrhage, rapid cognitive impairment, or stereotyped neurological deficits. On a non-enhanced brain CT scan, classic features account for posterior leucoaraiosis, hydrocephalus, or lacunar lesions. On MRI, microhemorrhages or superficial siderosis on gradient-echo T2 magnetic resonance imaging are signs of CAA. There is a high risk of recurrent hemorrhage in patients with intracerebral hemorrhage secondary to cerebral amyloid angiopathy, above that expected with non-lobar hemorrhages. Balancing the risk of ischemic stroke and hemorrhagic stroke in patients with coexisting AF and CAA remains a challenging area and requires careful consideration on a case-by-case basis.

Anticoagulant and antithrombotic agents should be avoided in cases of cerebral amyloid angiopathy, unless the risk of ischemic stroke outweighs the high risk of hemorrhagic stroke.

A brain MRI in patients at high risk for CAA complications could allow the treatment with the best safety profile to be chosen.

Left-atrial appendage occlusion may provide a safe alternative for reducing the risk of ischemic stroke in atrial fibrillation for individuals in whom anticoagulation is contraindicated.

Future research must consider the underlying pathogenesis of the CAA-related ICH for better risk prediction and to guide treatment.

## Figures and Tables

**Figure 1 jcm-12-07704-f001:**
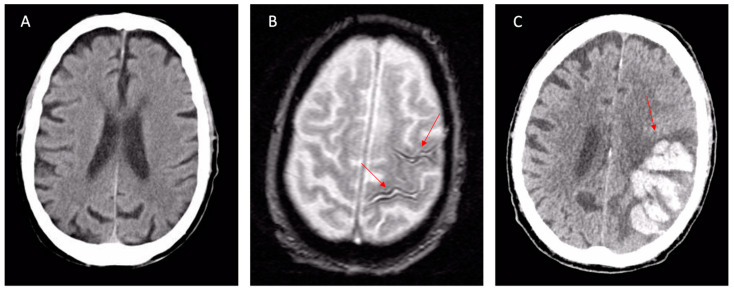
Final diagnoses of probable CAA according to the Boston Criteria 2.0. 84-year-old patient presenting with a TFNE (transient right hemiparesis and loss of speech for several minutes) and first detection of atrial fibrillation. (**A**) Apparently normal NECT. Oral anticoagulation was started for the secondary prevention of cardioembolism (the TFNE was misinterpreted as cardioembolic TIA). (**B**) MRI scan 10 days after the presentation showed signal loss in the left pre-central and central sulcus in T2* sequences (arrows), corresponding to “old” superficial siderosis. Oral anticoagulation was not interrupted based on CHA2DS2-VASc/HASBLED calculation. (**C**) Evolution with parenchymal left hematoma after few days (arrow). Abbreviations: NCET, non-enhanced computed tomography; CAA, cerebral amyloid angiopathy; TFNEs, transient focal neurological episodes; TIA, transient ischemic attack.

**Figure 2 jcm-12-07704-f002:**
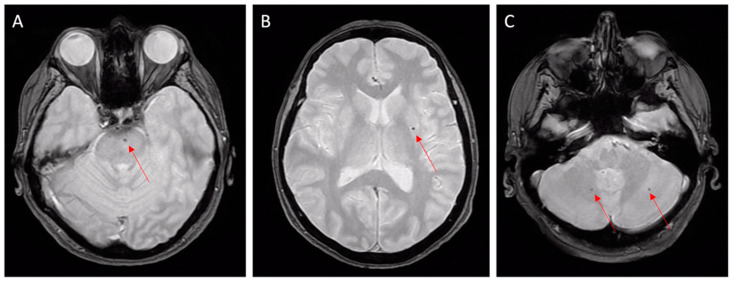
Hypertensive microangiopathy. Brain MRI, axial T2 gradient-echo images. Red arrows show microbleeds in the brain stem (**A**), left basal ganglia (**B**), and cerebellum (**C**). Deep localizations are typical features of hypertensive microbleeds.

**Figure 3 jcm-12-07704-f003:**
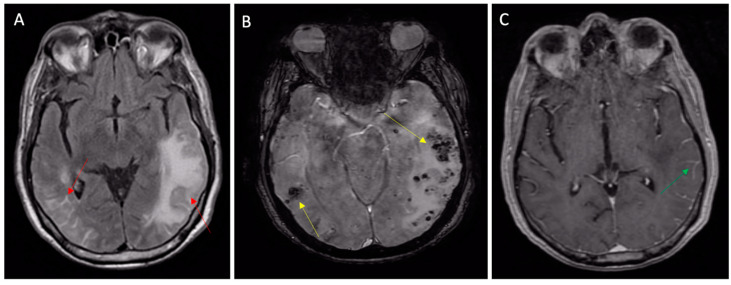
Inflammatory CAA presenting with TIA-MIMIC (transient aphasia). (**A**) MRI FLAIR sequences showing vasogenic edema (left arrow) and leptomeningeal inflammatory hyperintensities (right arrow); (**B**) susceptibility-weighted imaging MRI showing subcortical microbleeds (yellow arrows) and (**C**) leptomeningeal post contrast enhancement (green arrow) on T1 sequences.

**Figure 4 jcm-12-07704-f004:**
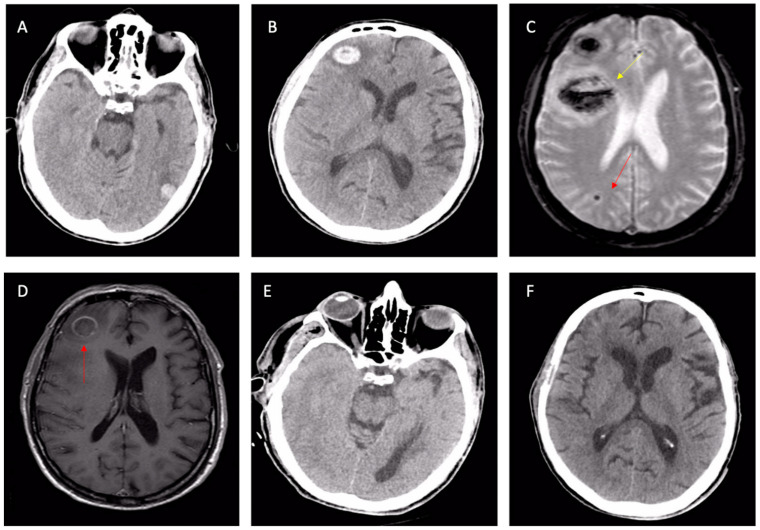
Cerebral involvement from confirmed bacterial endocarditis. (**A**,**B**) NECT showing left-temporal and right-frontal atypical hematomas linked with suspected CAA or cavernomas, according to radiologists. Same patient: second-line T2* brain MRI (**C**) with a further frontal lesion (yellow arrow) and subcortical microbleed (red arrow); the first two lesions are shown with signal loss in T2* sequences. The T1 post-contrast MRI (**D**) with ring enhancement (red arrow) raised the suspicion of an infective lesion. Control NECT after 3 weeks of antibiotic therapy (**E**,**F**): the complete disappearance of both frontal and temporal lesions, confirming cerebral emboli from endocarditis. Abbreviations: NCET, non-enhanced computed tomography; CAA, cerebral amyloid angiopathy.

**Table 1 jcm-12-07704-t001:** Clinical presentation of CAA.

Lobar ICH	Sudden Onset of Headache, Focal Neurological Signs, Seizures, or Altered State of Consciousness.
Cognitive impairment	Cognitive impairment not otherwise explained. Three possible variants: gradual, stepped, or rapidly progressive (expression of a greater inflammatory component).
Inflammatory forms	Subacute or rapidly progressive cognitive decline; possible epileptic manifestations. Susceptible to anti-inflammatory therapy response. Sometimes associated with TIA-LIKE presentations.
TFNEs (transient focal neurological episodes)	Recurrent and often stereotyped episodes of focal deficits (paresthesia, hyposthenia, aphasia) of variable duration between a few seconds and a few minutes. They often represent the clinical manifestation of subarachnoid hemorrhages. Brain CT can be negative. They can be interpreted as TIA and they could lead to the dangerous prescription of anticoagulant or antiplatelet therapy.

CT, computed tomography; TIA, transient ischemic attack.

**Table 2 jcm-12-07704-t002:** The Boston criteria version 2.0 for cerebral amyloid angiopathy [43]. Adapted with permission from [43] with permission of Elsevier. License Number 5687810797847, 14 December 2023.

CAA Probability	Diagnostic Criteria
Definite CAA	Full brain post-mortem examination demonstrating: spontaneous intracerebral hemorrhage, transient focal neurological episodes, convexity subarachnoid hemorrhage, or cognitive impairment or dementia severe CAA with vasculopathy absence of other diagnostic lesions
Probable CAA with supporting pathology	Clinical data and pathological tissue demonstrating: presentation with spontaneous intracerebral hemorrhage, transient focal neurological episodes, convexity subarachnoid hemorrhage, or cognitive impairment or dementia some degree of CAA in specimen absence of other diagnostic lesions
Probable CAA	For patients aged 50 years and older, clinical data and pathological tissue demonstrating: Presentation with spontaneous intracerebral hemorrhage, transient focal neurological episodes, or cognitive impairment or dementia At least two of the following strictly lobar hemorrhagic lesions on a T2*-weighted MRI, in any combination: - intracerebral hemorrhage, cerebral microbleeds, or foci of cortical superficial siderosis (multiple distinct foci are counted as independent hemorrhagic lesions) - convexity subarachnoid hemorrhage (multiple distinct foci are counted as independent hemorrhagic lesions) or One lobar hemorrhagic lesion plus one white matter feature (severe perivascular spaces in the centrum semiovale or white matter hyperintensities in a multispot pattern) Absence of any deep hemorrhagic lesions on a T2*weighted MRI (hemorrhagic lesion in cerebellum not counted as either a lobar or deep hemorrhagic lesion) Absence of any other cause of hemorrhagic lesions
Possible CAA	For patients aged 50 years and older, clinical data and pathological tissue demonstrating: Presentation with spontaneous intracerebral hemorrhage, transient focal neurological episodes, or cognitive impairment or dementia Absence of any other cause of hemorrhagic lesions One strictly lobar hemorrhagic lesion on a T2*-weighted MRI: intracerebral hemorrhage, cerebral microbleeds, or foci of cortical superficial siderosis or convexity subarachnoid hemorrhage or One white matter feature (severe perivascular spaces in the centrum semiovale or white matter hyperintensities in a multispot pattern) Absence of any deep hemorrhagic lesions on a T2*weighted MRI (hemorrhagic lesion in cerebellum not counted as either a lobar or deep hemorrhagic lesion) Absence of any other cause of hemorrhagic lesions

Abbreviations: CAA, cerebral amyloid angiopathy; MRI, magnetic resonance imaging.

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
