# Peer review of "When Atrial Fibrillation Meets Cerebral Amyloid Angiopathy: Current Evidence and Strategies"

_jcm, 2023, doi:10.3390/jcm12247704_

Round 1

Reviewer 1 Report

Comments and Suggestions for Authors

Dear Sir/Madam,

I had the opportunity to act as a reviewer on the recent submission by Merella et al. to the Journal of Clinical Medicine.

The authors present a comprehensive review regarding cerebral amyloid angiopathy and atrial fibrillation. 

The manuscript is well structured and written. However, following issues needs to be addressed: 

1.     Current evidence regarding catheter ablation (i.e., pulmonary vein isolation) in this special subset of patients.

2.     Please mind the correct verb conjugation in the title: “meets”

3.     Figure 1: correction of spelling needed, please also mind the abbreviations (what does “NECT stand for?)

Best regards,

Author Response

Revisor 1:

  1. Current evidence regarding catheter ablation (i.e., pulmonary vein isolation) in this special subset of patients.

We add a small paragraph about this issue (line 320)

  1. Please mind the correct verb conjugation in the title: “meets”

            We did.

  1. Figure 1: correction of spelling needed, please also mind the abbreviations (what does “NECT stand for?)

            We did.

Reviewer 2 Report

Comments and Suggestions for Authors

The authors present a review describing cerebral amyloid angiopathy (CAA) with focus on management of atrial fibrillation (AF) due to increased risk of intracranial bleeding. Though there are a number of previously published reviews focused on the same issue, the review is interesting and well-written. However, to my opinion a number of important recently published articles missed by the authors (DOI: 10.1093/brain/awad193, DOI: 10.3389/fcell.2023.1156970).

I would recommend to expand the part describing the pathogenesis of CAA (the authors paid a little attention to the mechanisms of CAA development) and add the section on the possible cardiac involvement in the development of the disease as AF might be a clinical presentation of cardiac amyloidosis.

Minor comments:

line 31 - "this results were....", should be these

lines 38,128 - should be CAA instead of CCA - misprinting

line 300 - should be AF instead of FA - misprinting

Comments on the Quality of English Language

The general quality is good, several misprintings (see comments above).

Author Response

Reviewer 2

However, to my opinion a number of important recently published articles missed by the authors (DOI: 10.1093/brain/awad193, DOI: 10.3389/fcell.2023.1156970).

We evaluated and inserted the recently published article.

I would recommend to expand the part describing the pathogenesis of CAA (the authors paid a little attention to the mechanisms of CAA development) and add the section on the possible cardiac involvement in the development of the disease as AF might be a clinical presentation of cardiac amyloidosis.

We have evaluated this possibility, but we do not agree for two reasons:

- first of all, there are already many publications that describe the pathogenetic mechanisms in depth;

-the purpose of our publication was to provide a practical guide to the cardiologist and the internal medicine specialist on what the disease is, when it should be suspected, how to diagnose it and how to treat any atrial fibrillation, when present.

Therefore, if deemed necessary, we will proceed to insert the requested section, otherwise we proceed with this setting.

line 31 - "this results were....", should be these; lines 38,128 - should be CAA instead of CCA – misprinting; line 300 - should be AF instead of FA - misprinting

We modified as suggested.

Reviewer 3 Report

Comments and Suggestions for Authors

This manuscript is well written and designed properly. However few issues might improve this article.

Please add some paragraph about mixed anticoagulation and antiplatelet treatment especially in patients with STEMI/NSTEMI, ecpesiallyin highe bleeding risk patients.

This study might be useful for authors to describe this topic

https://pubmed.ncbi.nlm.nih.gov/33708474/

Any suggested more ofte kidney function evaluation in patients with NOAC/DOAC and CAA? How do authors suggest to check thoses patients in case of bleeding risk?

Comments on the Quality of English Language

thank you for opportunity to cooperate

Author Response

Dear reviewer, we have evaluated this possibility, but the paper is focused on Atrial Fibrillation patients with concomitant cerebral amyloid angiopathy. Therefore, we preferred not to develop this theme.

Round 2

Reviewer 2 Report

Comments and Suggestions for Authors

I would like to thank the authors for their responds and corrections, I agree with their arguments.